# Combined Effects of Carotenoids and Polyphenols in Balancing the Response of Skin Cells to UV Irradiation

**DOI:** 10.3390/molecules26071931

**Published:** 2021-03-30

**Authors:** Glenda Calniquer, Marina Khanin, Hilla Ovadia, Karin Linnewiel-Hermoni, David Stepensky, Aviram Trachtenberg, Tanya Sedlov, Oleg Braverman, Joseph Levy, Yoav Sharoni

**Affiliations:** 1Department of Clinical Biochemistry and Pharmacology, Faculty of Health Sciences, Ben-Gurion University of the Negev, Beer-Sheva 84105, Israel; calnigue@post.bgu.ac.il (G.C.); hanin@bgu.ac.il (M.K.); ovadiahi@post.bgu.ac.il (H.O.); davidst@bgu.ac.il (D.S.); aviramtr@post.bgu.ac.il (A.T.); lyossi@bgu.ac.il (J.L.); 2Lycored, Secaucus, NJ 07094, USA; Karin.Hermoni@lycored.com; 3Lycored Ltd., Hebron Rd. P.O.B. 320, Beer Sheva 84102, Israel; Tanya.Sedlov@lycored.com (T.S.); Oleg.Braverman@lycored.com (O.B.)

**Keywords:** lycopene, carnosic acid, keratinocytes, interleukin-6, antioxidant response element/Nrf2, NFκB, MMP-1

## Abstract

Oral carotenoids and polyphenols have been suggested to induce photo-protective effects. The aim of the study was to test whether the combination of carotenoids and polyphenols produce greater protective effects from UV-induced damage to skin cells. Such damage is characterized by inflammation and oxidative stress; thus, the photo-protective effect can be partially explained by modulating the nuclear factor kappa B (NFκB) and antioxidant response element/Nrf2 (ARE/Nrf2) transcription systems, known as important regulators of these two processes. Indeed, it was found in keratinocytes that carotenoids and polyphenols inhibit UVB-induced NFκB activity and release of cytokine IL-6. A combination of tomato extract with rosemary extract inhibited UVB-induced release of IL-6 more than each of the compounds alone. Moreover, this combination synergistically activated ARE/Nrf2 transcription systems. Inflammatory cytokines such as IL-6 and TNFα induce the expression of matrix metalloproteinases (MMPs), which leads to collagen breakdown; thus, it is important to note that carnosic acid reduced TNFα-induced MMP-1 secretion from human dermal fibroblasts. The in vitro results suggest beneficial effects of phytonutrient combinations on skin health. To assure that clinical experiments to prove such effects in humans are feasible, the human bioavailability of carotenoids from tomato extract was tested, and nearly a twofold increase in their plasma concentrations was detected. This study demonstrates that carotenoids and polyphenols cooperate in balancing UV-induced skin cell damage, and suggests that NFκB and ARE/Nrf2 are involved in these effects.

## 1. Introduction

Exposure to sun irradiation causes skin damage that includes erythema (skin inflammation), premature skin aging, DNA damage, cell death, and skin cancer. Human intervention studies have shown that various carotenoids such as lycopene and β-carotene protect skin from UV-induced damage [1,2,3,4,5,6,7]. Polyphenols are another group of phytonutrients with similar action [8]. Photo-protective effects of carotenoids and polyphenols were attained also in in vitro studies. For example, lycopene, beta-carotene, and lutein reduced UV-induced lipid peroxidation in human dermal fibroblasts [9] and carnosic acid, a phenolic diterpene from rosemary, prevents UV-induced damage in human fibroblasts and keratinocytes [10]. A tenable view is that the beneficial effects of phytonutrient mixtures reside, at least partly, on the complementary and overlapping mechanisms of action of these nutrients on several cellular pathways. Modulation of transcription and gene expression has been found to play a significant role in the effect of phytonutrients on various cellular processes [11] including the antioxidant defense mechanism and inflammatory processes.

Exposure to the sun is followed by the development of inflammation which produces erythema. This is accompanied by increased blood flow to the affected skin area and activation of the nuclear factor kappa B (NFκB) transcription system in the immune cells, as well as in keratinocytes and dermal fibroblasts [12]. NFκB induces the synthesis and secretion of inflammatory cytokines such as IL-6 and TNF-α [12,13,14,15]. These cytokines, in turn, activate the synthesis and secretion of proteases that degrade skin collagen such as matrix metalloproteinases (MMPs) [2,13,16,17,18]. Thus, to determine if the combination concept is true also in photo-protection, the effect of carotenoids and polyphenols combinations was tested on UVB-induced cytokine secretion. Since NFκB is one of the mechanisms for increased cytokine secretion by UV-irradiation, the inhibition of NFκB by carotenoids and polyphenols may provide a mechanism for skin protection by phytonutrient combinations. Actually, several natural compound where shown to protect skin by inhibiting the NFκB pathway [13,19]. UV exposure causes the generation of reactive oxygen species (ROS) which contribute to skin damage [13,20]. A major mechanism in the cellular defense against oxidative or electrophilic stress is activation of the antioxidant response element/Nrf2 (ARE/Nrf2) transcription system, which controls the expression of genes whose protein products are involved in detoxification and the elimination of ROS [21].

The two transcription systems, ARE/Nrf2 and NFκB, are known to be modulated by phytonutrients [22,23,24]; however, they induce genes that have contrasting effects on the skin. The ARE/Nrf2 system induces genes of antioxidant and detoxifying enzymes that neutralize harmful molecules in the skin, whereas NFκB induces genes that are involved in inflammatory processes and that are deleterious to the skin [25]. This brought up the hypothesis that activation of ARE/Nrf2 and inhibition of NFκB by phytonutrient combinations will additively or even synergistically affect skin cells, which can improve skin health. Thus, the aim of this study was to compare the effects of carotenoids and polyphenol combinations on ARE/Nrf2 and NFκB transcriptional activity to the effects on cytokine and MMP secretion. Indeed, the main finding of the study was that the combination of carotenoids and polyphenols inhibited UVB-induced release of IL-6 in a cooperative way, reaching much stronger inhibition than each of the compounds alone. These combinations caused the opposite effect on ARE/Nrf2, where very strong synergistic activation was evident. The correlation between the effects of phytonutrients on the transcription systems (activation of Are/Nrf2 and inhibition of NFκB) and reduction in the damaging effects in the skin cells (IL-6 and MMP1 secretion) suggests a role for modulation of the transcriptional activities in balancing the response of skin cells to UV irradiation. Another important aspect of the study is the analysis of the high carotenoid blood concentrations that can be achieved with ingestion of tomato extract. This may have practical implications for the use of dietary intervention in improving human health.

## 2. Results

### 2.1. Carotenoids and Polyphenols Inhibit UVB-Induced Activation of the NFκB Transcription System

The NFκB transcription system is activated during inflammation, and UV irradiation is known to elicit skin inflammation. Indeed, in an NFκB reporter gene assay, it was found that UVB irradiation of keratinocytes activated this transcription system in a dose-dependent manner (Figure 1a). It is well documented from previous work in our laboratory that carotenoids and polyphenols inhibit this transcription system in several cellular experimental models [26]. Thus, their effects on NFκB transcriptional activity were evaluated. Lycopene, lutein, and curcumin significantly inhibited UVB-induced NFκB reporter activity, and carnosic acid inhibited it only slightly, but β-carotene did not show any effect (Figure 1b).

While intact, un-oxidized carotenoids can influence various signaling pathways [11], in our previous studies it was shown that, in fact, oxidized derivatives of carotenoids mediate at least some of these activities. For example, we recently observed that apo-carotenals are the active mediators in NFκB inhibition [26], as well as in ARE/Nrf2 induction [27]. Similarly, in the current study, the lycopene derivative 6,14′-diapocarotene-6,14′-dial (6,14′) inhibited NFκB activity to a greater extent than lycopene (Figure 1b).

### 2.2. Cooperative Inhibition of UVB-Induced IL-6 Secretion by Combinations of Carotenoids and Polyphenols

In immune cells, activation of the NFκB transcription system during inflammation induces the secretion of a family of cytokines including IL-6. This and other cytokines stimulate the inflammatory response. IL-6 is also secreted from epidermal cells during the inflammatory response induced by UV irradiation. The level of UVB-induced IL-6 was measured in the medium of irradiated HaCaT cells. IL-6 release was dependent on the irradiation dose, and the maximal effect was achieved at 30 mJ/cm^2^ (Figure 2a). The effect of carotenoids and polyphenols on the UVB-induced release of IL-6 was then examined (Figure 2b). The effect of carnosic acid and curcumin was dose dependent. However, only the highest concentration of lycopene inhibited IL-6 release from the HaCaT cells.

The inhibition of IL-6 release by the phytonutrients was significant only at high concentrations. Thus, it was tested if a greater inhibition would be achieved if combinations of nutrients were used instead of single compounds. When combining concentrations at the IC_50_ of the tested compounds, it is impossible to achieve more than an additive effect since such an effect will theoretically give 100% inhibition. Thus, to test for possible cooperative inhibition, concentrations of the different dietary compounds that are below the IC_50_ have to be used, and where the sum of the activities was below 50% inhibition. Indeed, the combination of carnosic acid and lycopene inhibited UVB-induced IL-6 release to a greater extent than the sum of the percent inhibition by each compound alone (Figure 2c). Then it was tested whether similar cooperative inhibition of UVB induced IL-6 release could be achieved with the carotenoid-rich Tomato Nutrient Complex (TNC) instead of purified lycopene. The combination of TNC with curcumin (Figure 2d) or carnosic acid (Figure 2e) resulted in a similar cooperative inhibition of IL-6 release with a statistically significant difference between the sums of the effects of the compounds alone as compared to their combination. The combination of TNC with rosemary extract containing carnosic acid (Figure 2f) showed a similar trend. However, because the effect of the rosemary extract alone was high, the combined effect was not significantly different from the sum of the effects.

### 2.3. Matrix Metalloproteinase-1 (MMP-1) Secretion from Human Dermal Fibroblasts Is Reduced by Carnosic Acid Treatment

In the results described above, it was shown that phytonutrient combinations inhibited UV-induced NFκB activity and reduced the secretion of cytokines that were induced by this transcription system. Inflammatory cytokines such as IL-6 and TNFα induce the expression of matrix metalloproteinases (MMPs) [28], which leads to the breakdown of collagen, the major structural component of the skin [29]. Thus, in normal human dermal fibroblasts (NHDFs), the effect of phytonutrients on cytokine-induced secretion of MMP-1 was checked. The pro-inflammatory cytokine, TNFα, increased the secretion of MMP-1 by about fivefold (Figure 3). Pre-treatment of the cells with carnosic acid resulted in a dose-dependent reduction in the MMP-1 protein level (Figure 3a). Treatment of the fibroblasts with rosemary extract containing the same concentrations of carnosic acid produced a similar reduction in the MMP-1 protein level (Figure 3b).

### 2.4. Synergistic Activation of the ARE/Nrf2 Transcription System in Skin Cells by Phytonutrient Combinations

The damaging effects of UV irradiation in skin may result from its direct effect on DNA and proteins in the cells, or through photochemical generation of singlet oxygen and other ROS, which increase oxidative stress in the cells, and may result in acute and chronic health effects on the skin. One of the cellular mechanisms that reduces oxidative stress is activation of the ARE/Nrf2 transcription system, which results in increased expression of antioxidant enzymes such as catalase and superoxide dismutase. We have previously found that carotenoids activate this transcription system [22] and determined the mechanism of this activation in cancer cells [27]. In the current study, the effect of TNC containing lycopene and rosemary extract containing carnosic acid on the activation of the ARE/Nrf2 transcription system in skin epidermal keratinocytes was tested.

Activation of this transcription system was measured in KERTr keratinocytes using a reporter gene assay. TNC containing 10 µM lycopene was combined with either purified carnosic acid (Figure 4a) or with rosemary extract containing carnosic acid (Figure 4b). The concentrations used in the combination experiments were such that the effect of each of the compounds tested alone, would be lower than 20% of the maximal activity achieved with the highest concentration of rosemary extract (containing 20 µM carnosic acid, about 200-fold). The combinations of TNC with carnosic acid or rosemary extract produced a statistically significant stronger activation of the ARE/Nrf2 reporter activity as compared to the compounds alone or to the sum of the effects of each compound alone. It is interesting to note that Lycoderm^TM^, which consists of TNC and rosemary extract, and was used in a concentration that gives 10 µM lycopene and 5 µM carnosic acid, increased the reporter activity similarly to that achieved with the combination of the same concentration of TNC and rosemary extract (Figure 4b, black column).

To test whether the interaction between these dietary compounds is synergistic, a combination index (CI) analysis was performed at various constant ratios between the compounds. The dose-effect curve of the combination of carnosic acid with TNC (Figure 4c) shows that the combined activity was greater at higher ratios of lycopene in TNC to carnosic acid. The combination index curve (Fa-CI plot, Figure 4d) shows a strong synergy between the two preparations with a lower CI (stronger synergy) at higher fractional effect. Figure 4e–g shows the average CI values for three different TNC concentrations (µM lycopene) at the three ratios shown in Figure 4c. This is an additional presentation of the strong synergy between TNC and carnosic acid in ARE/Nrf2 activation. A similar synergy is shown in Figure 4h–i for the combination of TNC with rosemary extract at a molar ratio of 4:1 (lycopene:carnosic acid). Of note, Lycoderm^TM^ containing TNC and rosemary extract at a molar ratio of 2:1 (lycopene:carnosic acid) exhibited even stronger synergy.

### 2.5. Bioavailability of Carotenoids from Lycored Tomato Nutrient Complex for Skin (Lycoderm^TM^)

The results of the above experiments suggest beneficial effects of phytonutrient combinations for the skin. However, these results were obtained in in vitro experiments; thus, there was a need to prove these effects in a human setting. To assure that such clinical experiments are feasible, the human bioavailability of carotenoids found in TNC for skin (Lycoderm^TM^) was analyzed. Twenty-three volunteers (aged 24 to 33, 10 men and 13 women) were recruited for the study, conducted as described in the Materials and Methods section. They consumed two capsules of Lycoderm^TM^ per day at two fat-containing meals. The amounts consumed per day of the main phytonutrients in the complex were 15 mg lycopene, 5.9 mg phytoene plus phytofluene, 0.9 mg β-carotene, and 4 mg carnosic acid (as rosemary extract). Blood withdrawal was done after a 12-h fast at time zero and at two and three weeks after beginning supplementation. The participants tolerated the supplement well, and no adverse effects were recorded. There was substantial between-subjects variability in the baseline concentrations of the studied phytonutrients (lycopene, phytoene, phytofluene, and β-carotene), and in the time course and maximal value of their concentrations following multiple oral administration of the supplement (shown for lycopene in Figure 5a; the numerical values of individual concentrations of all tested carotenoids are found in Appendix A). The baseline levels for lycopene ranged from 0.5 to 2.0 µM, and its maximal levels ranged from 0.6 to 3.2 µM. A statistically significant and linear increase in the average plasma lycopene, phytoene, and phytofluene was observed after two and three weeks of supplementation (Figure 5b), which for lycopene reached a maximal level of 1.9 ± 0.6 µM. Although the amount of β-carotene in the supplement was only 0.9 mg per day, a small but statistically-significant increase was evident at week three. Steady-state concentrations were still not attained after two weeks of treatment, and it is not clear if they were attained after three weeks. However, from similar experiments (Sharoni, Y., unpublished), we know that steady-state levels are usually achieved after three weeks or sooner. Carotenoid plasma concentrations were measured again, three and four weeks after the end of supplementation, and they returned to baseline values after four weeks.

## 3. Discussion

This study supports the role of dietary polyphenols and carotenoids in skin protection from UV-induced damage and proposes mechanisms for their action. Although in the current study only a limited number of phytonutrients were used, the results strengthen the hypothesis that the additive and synergistic effects of phytonutrients from fruits and vegetables are important for their potent antioxidant and anti-inflammatory activities, which protect skin from damage induced by sun exposure. We formerly obtained similar synergistic effects of phytonutrient combinations on the inhibition of osteoclast differentiation [30] and the inhibition of prostate cancer cell proliferation by various phytonutrient combinations [31].

As model nutrients for the carotenoids, purified lycopene or carotenoid-rich TNC were used. In addition to the main carotenoids of lycopene, phytoene, and phytofluene, this tomato extract contains other carotenoids, polyphenols, sterols, and vitamins (see Section 4.1). Carnosic acid, curcumin, and a rosemary extract containing mainly carnosic acid and carnosol represent the vast polyphenol family. Most of these nutrients were previously shown to prevent UV-induced skin damage [1,2,3,4,5,6,7,8,9,10]. To understand how phytonutrient combinations protect the skin, the effect of their combinations on several signaling components that are induced by UV irradiation was evaluated (see model in Scheme 1). This includes the NFκB transcription system upon which induction by UV leads to an increase in the level of various inflammatory cytokines, including IL-6 and TNFα [13,19] that enhance inflammatory responses such as erythema and sunburn [12]. Indeed, combinations of various carotenoids and polyphenols inhibit NFκB activity and inhibit the production of IL-6 in a cooperative manner, which results in activity that is higher than the sum of the individual activities. The attenuation of the TNFα-induced increase in the level of the metalloproteinase MMP-1 is another result of the phytonutrient protective activity along this inflammatory cascade. This reduction in MMP-1 activity may reduce extracellular matrix degradation [13], as suggested in Scheme 1. Contrary to the inhibition of the NFκB transcription system, the phytonutrient combinations resulted in a synergistic activation of the ARE/Nrf2 transcription system. UV-induced ROS generation is a major mechanism through which UV causes detrimental effects on skin [32] and, thus, activation of ARE/Nrf2 is one of the basic mechanisms to balance UV-induced skin damage, as was shown in a 3D skin model injected with Nrf2 over-expressing skin-derived precursor cells [33]. The inhibition of NFκB and activation of ARE/Nrf2 by various carotenoids and polyphenols was previously shown in several publications [22,26,34,35]. However, in most cases, the concentrations required for these effects were higher than those that can be achieved in human blood and tissues [30,36]. This discrepancy enhances the importance of the results presented in the current study, showing that phytonutrient combinations caused, in some cases, four to five times higher inhibition of IL-6 secretion and activation of ARE/Nrf2 activity than the activity of the single compounds. The ARE/Nrf2 mediated protection is probably due to a reduction in oxidative stress [33,37], and also because the ARE/Nrf2 pathway inhibits NFκB-mediated transcription [38,39,40] (see Scheme 1). This may occur by several mechanisms, e.g., one of the target proteins of Nrf2, heme oxygenase 1, binds to nuclear NFκB p65 and inhibits its activity [41], and Kelch-like ECH-associated protein 1 (KEAP1) which is the negative regulator of Nrf2, can interact with IκB kinase β (IKKβ), which is a positive regulator of NFκB. This interaction causes IKKβ degradation and suppression of NFκB [42]. Due to such interactions between ARE/Nrf2 and NFκB pathways, it is expected that combinations of carotenoids and polyphenols will synergistically prevent various manifestations of skin damage.

The mechanisms for the cooperative effects in the prevention of skin damage and the synergistic activation of ARE/Nrf2 are not completely clear, but there are several possible explanations that should be addressed in future studies. There are several pathways to activate NFκB and many proteins are involved in its regulation [43], which suggest that modulating the different pathways can lead to cooperative activity of several phytonutrients. In this study, only the NFκB pathway was addressed as one mechanism for the skin damaging effects of UV-irradiation. However, it is known that other pathways participate in this process such as mitogen-activated protein kinases (MAPKs), mammalian target of rapamycin complex 1 (mTORC1), phosphatidyl-inositol 3-kinases (PI3K), toll-like receptors, and possibly others [35]. Thus, it is suggested that different phytonutrients interfere with distinct pathways, resulting in a cumulative total effect. As to the synergistic activation of ARE/Nrf2, although the main mechanism for activation of this transcription system is through the interaction of electrophiles with the Keap1 protein, other mechanisms are also known [34]. Firstly, the transcription of ARE/Nrf2-regulated genes is induced by heterodimers of Nrf2 with other proteins, especially with small musculoaponeurotic fibrosarcoma proteins (sMafs). In addition, Nrf2 phosphorylation and acetylation in various amino acids can also affect transcriptional activity [34]. Thus, affecting the relative amount of sMafs and the degree of phosphorylation and acetylation by the various phytonutrients may contribute to the synergy.

The results of this study open new avenues for prevention of UV-induced skin damage. A nutritional approach has clear advantages over topical prevention with sunscreens, which are used mainly during vacation time, whereas about 70% of sun exposure occurs during regular daily activity [7]. However, a major limitation of the current study is the nature of in vitro experiments, which may not reflect the same results that may be obtained in people. Thus, achieving similar results in an in vivo setting is required. Therefore, human volunteers were exposed to orally-consumed carotenoids. Specifically, the bioavailability of the main carotenoids that are found in a relevant tomato extract-based preparation (lycopene, phytoene, and phytofluene) was analyzed. The results clearly demonstrate that the average plasma concentration of the major tomato carotenoids almost doubled. This large increase resulted in an average lycopene concentration of 1.9 ± 0.6 µM. It should be noted that this concentration was below that used for TNC in the various in vitro experiments (5–10 µM lycopene). This raises the question as to whether the in vitro results have any relevance to the human situation. To answer this, it must be realized that, in vivo, blood carotenoids are found mainly in lipoproteins such as chylomicrons during the absorptive phase, and in LDL and VLDL, in the post-absorptive phase [44]. The lipoproteins containing carotenoids enter the cells mainly by receptor-mediated processes, which are very efficient. The carotenoids in the current in vitro experiments were solubilized using THF, which results in a micro-emulsion rather than molecular solution [45]. Although the carotenoids in such preparations do enter the cells [45], it is assumed that this process is less efficient than the cellular accumulation of carotenoids from lipoproteins in vivo, and thus, higher concentrations of carotenoids are required in vitro to imitate the in vivo effects.

In previous studies [46], we found that with TNC supplementation for a similar, or even shorter, period of time than in the current study, the increase in the plasma lycopene level was similar to that in skin. In a recent human study, supplementing the same TNC for skin (Lycoderm^TM^) at 7.5 mg/day (half of the amount supplemented in the current study), an increase of about 30% in skin carotenoid levels was detected [47], which suggests that it is possible to obtain various effects of carotenoid supplementation in human skin. Indeed, in that human study, a significant reduction in the appearance of lines and wrinkles in subjects using the supplement was detected, jointly with subjective tangible benefits to the skin. In another clinical study, conducted by Krutmann et al. [48], the results show that treatment with a combination of TNC and rosemary extract reduced UV-induced skin inflammatory response. These results agree with the in vitro results presented here.

## 4. Materials and Methods

### 4.1. Materials

Crystalline lycopene preparations purified from tomato extract (>97%), carotenoid-rich Tomato Nutrient Complex (TNC), rosemary extract, and Lycoderm^TM^, which is a standardized combination of TNC and rosemary extract (see details in Section 2.4 and Section 2.5) were a gift of Lycored Ltd., Beer Sheva, Israel. The TNC contained 6% lycopene, other tomato carotenoids (phytoene 1.5%, phytofluene 0.6%, beta-carotene 0.4%), and additional fat-soluble tomato components such as phytosterols (1.7%) and tocopherols (2.0%). The main polyphenols in the rosemary extract were carnosic acid (20.2%) and carnosol (2.5%). β-carotene was a gift of DSM (Basel, Switzerland). Crystalline curcumin (>95%) was purchased from Cayman Chemicals (Ann Arbor, MI, USA). Carnosic acid (93–97%) was purchased from Alexis Biochemicals (Switzerland), and tetrahydrofuran (THF), containing 0.025% butylated hydroxytoluene as an antioxidant, was purchased from Aldrich (Milwaukee, WI, USA). Dulbecco’s modified Eagle’s medium (DMEM), and phosphate buffered saline (PBS) were purchased from Biological Industries (Beth Haemek, Israel). Fetal calf serum (FCS) was purchased from Gibco (Grand Island, NY, USA). Carotenoids were dissolved in THF, solubilized in cell culture medium and their final concentration was measured, as described previously [27,45,49]. Acrodisc^®^ syringe filters 0.8/0.2 µm (PALL Corporation, Port Washington, NY, USA) were used to remove precipitates.

### 4.2. Cell Culture and Treatment

HaCaT keratinocytes and KERTr keratinocytes were purchased from American Type Culture Collection (Manassas, VA, USA). The HaCaT cells were grown in DMEM medium supplemented with penicillin (100 U/mL), streptomycin (0.1 mg/mL), glutamine (2 mM), and 10% FCS. KERTr cells were grown in keratinocyte serum-free medium (Gibco) supplemented with bovine pituitary extract and Epidermal growth factor (EGF). Normal human dermal fibroblasts (NHDF) were purchased from PromoCell GmbH (Heidelberg, Germany). The cells were grown in fibroblast growth medium 2 (PromoCell), according to the manufacturer’s instructions.

### 4.3. Reporter Constructs

NFκB-SEAP reporter construct was purchased from Clontech Laboratories, Inc. (Palo Alto, CA, USA). 4xARE reporter construct was kindly provided by Dr. M. Hannink (University of Missouri–Columbia, Columbia, MO, USA). CMV-luciferase and renilla luciferase (*p*-RL-null) expression vectors, which served as internal transfection standards, were purchased from Promega (Madison, WI, USA).

### 4.4. Transient Transfection and Reporter Gene Assay

Cells were transfected using jetPEI reagent (Polyplus Transfection, Illkrich, France) in 24-well plates. HaCaT cells (80,000 cells per well) were transfected with 0.2 µg NFκB-SEAP reporter construct and 0.05 µg normalizing plasmid. The ratio of DNA to jetPEI was 1:5. KERTr keratinocytes were transfected with 0.2 µg 4xARE reporter construct and 0.1 µg normalizing plasmid. Cells were seeded in culture media containing 3% FCS. The next day, cells were rinsed once with the appropriate culture medium, followed by addition of 0.45 mL of medium and 50 µL of DNA mixed with jetPEI. Cells were then incubated for 4–6 h at 37 °C. For ARE/Nrf2 reporter activity, the medium was replaced with one supplemented with 3% FCS plus the test compounds, and cells were incubated for an additional 16–20 h. The reporter activity was measured in cell extracts and was normalized to renilla luciferase using a dual luciferase reporter assay system (Promega, Madison, WI, USA), according to the manufacturer’s instructions. For NFκB reporter activity measurements, the medium was replaced with PBS, and the cells were irradiated with a UVB lamp (302 nm; Ultra-Violet Products, Ltd., Upland, CA, USA). The intensity of irradiation was measured with a VLX-3W sensor (VilberLourmat Deutschland GmbH, Eberhardzell, Germany). Doses of 10–60 mJ/cm^2^ were used, and are indicated in the legends of relevant figures. The PBS was replaced with medium containing 3% FCS plus the test compounds, and the medium was collected after 6–9 h. Secreted alkaline phosphatase (SEAP) activity was measured in the culture media using a Great EscAPe™ SEAP chemiluminescence kit (Clontech, Mountain View, CA, USA), according to the manufacturer’s instructions. Luciferase activity was measured in cell extracts by a LUC assay kit (Promega, Madison, WI, USA), and was used to normalize the SEAP results. All luminescence measurements were performed in a Turner Biosystems luminometer (Sunnyvale, CA, USA). Of note, the transfection level in different experiments, as well as other parameters such as cell batch and passage, were variable. In combination experiments, comparisons between the different compounds in the same experiment were reproducible; however, basal reporter activity, as well as the fold induction, varied between experiments.

### 4.5. Measurement of IL-6 Levels

HaCaT cells were seeded in 24-well plates (120,000 cells/well) in DMEM medium supplemented with 10% FCS. The medium was changed to 3% FCS containing the different dietary compounds at the indicated concentrations. A day later, the medium was replaced with PBS, and the cells were irradiated with a UVB lamp, as described above. The PBS was replaced with medium containing 3% FCS plus the test compounds, and the medium was collected after 6 h. IL-6 in the medium was measured using a human IL-6 ELISA MAX™ Deluxe Kit (Biolegend, San Diego, CA, USA).

### 4.6. Measurement of Matrix Metalloproteinase-1 (MMP-1)

NHDF fibroblasts were seeded in 96-well plates (4500 cells/well) in DMEM medium supplemented with 3% FCS. The medium was changed to one containing the different dietary compounds at the indicated concentrations for one day. TNFα (1.5 ng/mL) was then added, and the medium was collected after 24 h. The amount of MMP-1 protein in the medium was measured using a human total MMP-1 ELISA DuoSet (R&D systems, Minneapolis, MN, USA).

### 4.7. Bioavailability of Carotenoid Supplements

The Ethics Committee of the Soroka University Medical Center approved the study (ethical approval code 0012-15-SOR). The volunteers signed an informed consent form, and were instructed to maintain their dietary routine. After withdrawal of 11 mL of blood at time zero, they received the supplement capsules and consumed two capsules daily at two separate meals. Additional blood samples were collected at two and three weeks after the beginning of the study. Subjects’ adherence to the experimental protocol was assessed by counting the unconsumed pills. Plasma was separated from the collected blood samples and was kept frozen until the end of the study. All plasma samples from an individual volunteer were processed together to minimize between-analysis variability. Plasma samples (0.5 mL) were mixed with an equal volume of ethanol and with astaxanthin as an internal standard, and were extracted by hexane/dichloromethane (4/1), as described before [50]. The extract was collected, evaporated, and re-dissolved in 0.2 mL 2-propanol. The samples thus obtained were analyzed using a Dionex Ultimate 3000 UHPLC system including quaternary pump, degasser, column oven, autosampler, and PDA detector. Gradient elution was applied on a C18 reverse-phase HPLC column in the multi-wavelength mode. Mobile phase A: acetonitrile:methanol:ammonium acetate buffer = 25:25:50. Mobile phase B: acetonitrile:methanol:dichloromethane:hexane = 70:25:2.5:2.5. Gradient: 0 min—A 100%, 3.8 min—B 100%, 9.7 min—B 100%, 9.8 min—A 100%, 14.0 min—A 100%. The reported lycopene concentrations were calculated from the sum of the lycopene isomers’ peak area.

### 4.8. Statistical Analysis

All experiments were repeated at least twice, usually in triplicate, as detailed in the legend of each figure. Most experiments were repeated 4–8 times, but some specific conditions were repeated up to 20 times. The significance of the differences between the means of the various subgroups was assessed by a two-tailed Student’s t-test using a Microsoft Excel program. *p* ≤ 0.05 was considered statistically significant. To check for the significance of the cooperative effects of combinations of various compounds, the sum of the effects of each compound alone was calculated and compared to the effect of the combination using a Student’s t-test, as described above. The interaction between the various dietary compounds in inducing ARE/Nrf2 activity was assessed by combination index (CI) analysis using Calcusyn version 2.1, (BIOSOFT, Cambridge, Great Britain). The CI values were calculated based on the fold induction of ARE/Nrf2 (fraction affected) by each agent individually and by the combinations at constant ratios. CI values of <1, 1, and >1 represent synergism, additivity, and antagonism, respectively. In the bioavailability study, the differences in carotenoid concentrations between the various time-points was analyzed using a repeated measurements ANOVA with Tukey’s post-test.

## 5. Conclusions

Dietary carotenoids and polyphenols cooperate in balancing UV-induced skin cell damage and promote skin resilience to challenge. The NFκB and ARE/Nrf2 transcription systems are likely to be involved in these effects. The additive or even synergistic enhancement of ARE/Nrf2 activity and inhibition of IL-6 secretion by phytonutrient combinations support the use of mixtures of carotenoids and polyphenols to protect from sun-induced damage to skin. Thus, it is suggested that a diet, or supplements, containing combinations of various phytonutrients can improve skin health and appearance. Certainly, to protect human skin, the phytonutrients should be well absorbed from the diet. This was directly shown in the current study for some carotenoids, which reached steadily high plasma concentrations.

## Data Availability

The data presented in this study are available in the current manuscript and in the Appendix A.

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
