# Peer review of "Combined Effects of Carotenoids and Polyphenols in Balancing the Response of Skin Cells to UV Irradiation"

_molecules, 2021, doi:10.3390/molecules26071931_

Round 1
Reviewer 1 Report
The authors presented original research dealing with the effects of carotenoids and polyphenols in the response of skin cells to UV irradiation. It is known that carotenoids, as well as phenolic compounds, induce photo-protective effects. The authors utilize several methods to acknowledge their hypothesis and checked the effect of among others lycopene, curcumin, and carnosic acid, as well as their combinations and carotenoid-rich tomato nutrient complex. The research value of the manuscript is also increased by
testing the bioavailability of Lycoderm in a human setting with 23 volunteers.
In my opinion, the research was designed appropriate, methods were adequately described and only several suggestions were provided below.
Specific comments:
- Figure 5A is illegible. I propose to present the results as the table instead of the figure. Add numerical values into the table and then it may appear in the supplementary materials.
- The manuscript lacks in-depth discussion.
Minor comments:
- MMP and NFƙB should be defined in the abstract.
- The abstract is too long. It should be 200 words maximum.
- The conclusion should be rephrased and more information and some perspectives should be provided.
- The reference list should be improved.
Author Response
Specific comments:
- Figure 5A is illegible. I propose to present the results as the table instead of the figure. Add numerical values into the table and then it may appear in the supplementary materials.
Figure 5a is presented in order to show the large variability among the study participants in the baseline lycopene concentrations and in the increase in its plasma concentrations during the study. This information which is easily perceived from the figure, would not be easily apparent if the results are presented only in tables. We agree that adding the numerical values of all tested carotenoids does contribute greatly to the manuscript and thus, all numerical values are added in 4 tables as Supplementary Materials. A similar design was suggested also by Reviewer 3. A sentence referring to the Tables in the supplementary material was added in lines 282-283.
- The manuscript lacks in-depth discussion.
Several paragraphs were added to the discussion to address the following issues: The difference between the phytonutrient concentration used in the in-vitro studies and their human plasma concentrations which suggest the use of combinations (lines 342-350); the interactions between NFkB and Nrf2 pathways (lines 352-358); and possible mechanisms for the synergy (lines 363-381). In addition, several sentences were added to better explain other issues that were already present in the discussion. (Reference to line numbers here and thereafter are to the line numbers in the marked version so that the relevant changes can be easily identified by the red color of the text).
Minor comments:
- MMP and NFƙB should be defined in the abstract. The abbreviations were defined.
- The abstract is too long. It should be 200 words maximum.
According to comments of Reviewers 1 and 3, more information had to be added to the abstract. To compensate for these additions, several phrases were removed and after condensing the abstract, it is still 230 words. However, the assistant editor assigned to this manuscript, suggested that the abstract is not strictly limited to 200 words.
- The conclusion should be rephrased and more information and some perspectives should be provided.
Several sentences were added to the conclusion to improve it and to give better perspective, as suggested by all reviewers.
- The reference list should be improved.
Some older and less relevant references were removed and replaced by more appropriate ones. 17 new references were added—some into the original text, and mostly to the new added text. Most of these new references are more recent.
Reviewer 2 Report
Dear authors,
In the manuscript by Glenda Calniquer et al., you describe the combined effects of carotenoids and polyphenols in balancing UV-induced skin cell damage. I consider this work quite interesting, but I think it deserves another little effort to get better. My comments are as follows:
- In the introduction, you should better explain the novelty of the work. What does this research add to the state of the art?
- In this study, you used a limited number of phytonutrients. You should test more type of phytonutrients to strengthen your results.
- The Discussion session is poor. You should better discuss the obtained results aiming to highlight the strength of the work and the novelty of the obtained results.
- Your conclusions are poor. The conclusions are too generic. You should rewrite them.
Author Response
I consider this work quite interesting, but I think it deserves another little effort to get better. My comments are as follows:
- In the introduction, you should better explain the novelty of the work. What does this research add to the state of the art?
More information was added to the introduction in order to better explain the background of the study (lines 45-47 and 61-67) and to better explain the novelty of the work (lines 82-86 and 90-93). More references were added, and some were replaced with newer and more appropriate ones.
- In this study, you used a limited number of phytonutrients. You should test more type of phytonutrients to strengthen your results.
We agree that showing results with more phytonutrients would strengthen the results, but we believe that it will not change the concepts presented in this manuscript. Since this cannot be achieved in the short time allocated for the revision, we think that this should be left for future studies by us or by other researchers inspired by this study.
- The Discussion session is poor. You should better discuss the obtained results aiming to highlight the strength of the work and the novelty of the obtained results.
- Your conclusions are poor. The conclusions are too generic. You should rewrite them.
On points 3 and 4, see response to Reviewer 1.
Reviewer 3 Report
The manuscript titled “Combined effects of carotenoids and polyphenols in balancing the response of skin cells to UV irradiation” presented by Calniquer and colleagues have reported an interesting work about the effects of natural compounds in skin disease.
The manuscript is well written and presented.
However, there are some points to be clarified (reported below), which should be revised before the publication of the manuscript.
- Line 28 and in the entire manuscript: Please, put “in-vitro” and “in-vivo” in italic form.
- Line 17, line 19, line 30, line 32 and in the entire manuscript: Please, avoid to use personal form in scientific paper, as “our aim”, “we aimed”, “we checked”, “we demonstrate”, ecc.. Please, rephase these, for example: “In conclusion, the results obtained in this work have shown that carotenoids and polyphenols cooperate…” etc.
- Line 249: Section 2.5 Bioavailability of carotenoids from Lycored Tomato Nutrient Complex for skin (LycodermTM). In this section the authors have reported the in-vivo studies conduced on healthy volunteers. Please, include in this section a Table with quantitative data observed in plasma samples.
- Line 475: Conclusions. This section should be expanded.
Author Response
There are some points to be clarified (reported below), which should be revised before the publication of the manuscript.
- Line 28 and in the entire manuscript: Please, put “in-vitro” and “in-vivo” in italic form.
Changed to italic form
- Line 17, line 19, line 30, line 32 and in the entire manuscript: Please, avoid to use personal form in scientific paper, as “our aim”, “we aimed”, “we checked”, “we demonstrate”, ecc.. Please, rephrase these, for example: “In conclusion, the results obtained in this work have shown that carotenoids and polyphenols cooperate…” etc.
Most occurrences of "we" and "our" were rephrased. The ones that are left, should remain.
- Line 249: Section 2.5 Bioavailability of carotenoids from Lycored Tomato Nutrient Complex for skin (LycodermTM). In this section the authors have reported the in-vivostudies conducted on healthy volunteers. Please, include in this section a Table with quantitative data observed in plasma samples.
- Line 475: Conclusions. This section should be expanded.
See response to Reviewer 1 on the last two issues.